# Fixed Yellow-to-Blue Intensity Ratio of Dy^3+^ in KY(CO_3_)_2_ Host for Emission Color Tuning

**DOI:** 10.3390/ma17061438

**Published:** 2024-03-21

**Authors:** Lei Huang, Jian Qian, Shijian Sun, Dechuan Li

**Affiliations:** 1School of Physics and Electronic Information, Huaibei Normal University, Huaibei 235000, China; hbnu991210@163.com (L.H.); qianjianwyyx@163.com (J.Q.); sunsj_0105@163.com (S.S.); 2Anhui Province Key Laboratory of Pollutant Sensitive Materials and Environmental Remediation, Huaibei 235000, China

**Keywords:** Dy^3+^, color tuning, intensity ratio, phosphors

## Abstract

Dy^3+^, Ce^3+^ co-doped KY(CO_3_)_2_ phosphors with a monoclinic structure were synthesized using the hydrothermal method to create a fixed yellow-to-blue ratio emission. The [YO_8_] polyhedron, consisting of a Y atom and eight oxygen atoms, forms a relatively independent microstructure within the KY(CO_3_)_2_ host. Y^3+^ ions are partially replaced by Ce^3+^ or Dy^3+^ ions to construct the [CeO_8_] or [DyO_8_] polyhedral fluorescence emission unit. The spectral measurements indicate that Ce^3+^ and Dy^3+^ can maintain relatively independent fluorescence emission characteristics in the KY(CO_3_)_2_ host. The yellow-to-blue intensity ratio of Dy^3+^ remains close to 1 and does not change with the variation in the doping concentration of KY(CO_3_)_2_:Dy^3+^ and KY(CO_3_)_2_:Dy^3+^,Ce^3+^ phosphors. When Ce^3+^ and Dy^3+^ are co-doped with KY(CO_3_)_2_, the emission intensities of Dy^3+^ under 339 nm and 365 nm excitation increase by 8.43 and 2.32 times, respectively, through resonance energy transfer and cross-relaxation. All Ce^3+^-doped KY(CO_3_)_2_:Dy^3+^ phosphors can emit white light. Among them, the emitted light of KY(CO_3_)_2_:3%Dy^3+^,5%Ce^3+^ is closest to standard daylight. Therefore, a stable [YO_8_] polyhedral structure can be used to achieve more color tuning of light.

## 1. Introduction

Dy^3+^ is an important rare earth luminescent ion with strong blue and yellow emission [1,2]. Tunable emission properties are widely used in the fields of solid-state lighting [3,4,5], temperature sensing [6,7,8], gamma detection [9], and anti-counterfeiting [10,11]. The intense emission of two visible lights is attributed to the energy level transitions from ^4^F_9/2_ to ^6^H_15/2_ and ^6^H_13/2_ [3]. The ^4^F_9/2_ → ^6^H_15/2_ transition belongs to the magnetic dipole transition, whereas the ^4^F_9/2_ → ^6^H_13/2_ transition belongs to the hypersensitive electric dipole transition, which is greatly affected by the crystal field environment [12]. The electric field environment around Dy^3+^ can be modified by doping with different concentrations of ions in the host. The higher the asymmetry is, the greater the yellow emission intensity [13,14]. An asymmetric crystal field changes the yellow-to-blue intensity ratio in the Dy^3+^ emission spectrum, resulting in an unpredictable variety of luminescent color. However, the doping concentration should not be excessive for Dy^3+^. An excessive concentration in the crystal lattice could reduce the distance between two Dy^3+^ ions and promote cross-relaxation and resonance energy transfer [14,15]. Consequently, the emission intensity is weakened due to concentration quenching [16,17,18]. Therefore, it is essential to investigate an asymmetric fluorescent material with a relatively fixed intensity ratio for intense emission [19].

Asymmetric microstructures are widely used for the emission of Dy^3+^ [20,21]. A phosphor with a tetragon tungsten bronze structure is used to produce stable blue and yellow light emissions [12]. However, the yellow/blue ratio is limited due to the relatively balanced tendency of Dy^3+^ in both pentagonal-A and square-B sites of the PbTa_2_O_6_ crystal structure. The yellow emission of ^4^F_9/2_ → ^6^H_13/2_ corresponds to an electric dipole transition, which is sensitive to the asymmetric arrangement of Dy^3+^. The degree of asymmetry in the Dy^3+^ environment is often described by the yellow-to-blue intensity ratio (Y/B) [19]. The yellow light intensity is highly uncertain in differently doped hosts. The Y/B ratios are larger than 1 in Dy^3+^-doped Bi_2_MoO_6_ [15], K_3_YB_6_O_12_ [16], and YGdPO_4_ [20] and less than 1 in Y_4_AI_2_O_9_ [5], PbTa_2_O_6_ [12], and YAI_5_O_12_ [22]. Moreover, the value of Y/B does not change synchronously with the concentration of Dy^3+^ [23]. Variation in the Y/B value in the Dy^3+^ emission spectrum may induce uncertainty in the luminescence color. Therefore, a stable crystal field environment with high emission intensity is particularly important for the single luminescent color of Dy^3+^. A host with a polyhedral structure is one of the good options.

As one of the fluorescent crystals, the monoclinic KY(CO_3_)_2_ is a good optical material with high optical transmittance [24]. In the [Y(CO_3_)_2_]^−^ framework of the KY(CO_3_)_2_ host, Y^3+^ is coordinated to eight O^2−^ to form a relatively independent [YO_8_] polyhedron. The Y^3+^ in this polyhedron can be replaced by other rare earth ions to achieve a stable luminescent emission [25]. The radius of Dy^3+^ ions is close to Y^3+^. The independent crystal field environment in the [DyO_8_] polyhedron ensures stable emission of Dy^3+^ in yellow and blue light, which means that the light color in KY(CO_3_)_2_:Dy^3+^ is relatively stable. In this work, Dy^3+^ and Ce^3+^ co-doped KY(CO_3_)_2_ are prepared, and the structures, morphologies, luminescence, decay curves, and color tuning are discussed.

## 2. Materials and Methods

A series of KY_1-*x*-*y*_(CO_3_)_2_:*x*%Dy^3+^,*y*%Ce^3+^ (KYC:*x*%Dy^3+^,*y*%Ce^3+^) phosphors were prepared by the hydrothermal method. High-purity Y(NO_3_)_3_·H_2_O (99.99%), Dy(NO_3_)_3_·H_2_O (99.99%), and Ce(NO_3_)_3_·H_2_O (99.99%) were used as raw materials for the chemical reactions (Shanghai Xianding Biotechnology Co., Ltd., Shanghai, China). The detailed synthesis process has been described in the literature [25]. First, all rare earth nitrates were weighed according to the stoichiometric ratio and dissolved in 5 mL deionized water. Second, the nitrate solution was added dropwise into a vigorously stirred 25 mL K_2_CO_3_ solution (0.55 mol/L). Third, the pH value of the nitrate mixed solution was adjusted to 9.5 using dilute nitric acid. After 30 min of vigorous stirring, the mixed solution was transferred to a 50 mL reactor and heated to 200 °C for 12 h. The reaction precipitate was filtered and washed with deionized water, and the final fluorescent powder was obtained.

The crystal structures of KYC:*x*%Dy^3+^,*y*%Ce^3+^ were characterized with an X-ray diffractometer. Lattice parameters were fitted by Rietveld refinement using FullProf software (5.10). The morphologies of grains and the types of elements were analyzed with cold-field emission scanning electron microscopy (Regulus 8220, Hitachi High-Tech Co., Tokyo, Japan). The luminescent properties of KYC:*x*%Dy^3+^,*y*%Ce^3+^ were detected with an FLS920 fluorescence spectrophotometer equipped with a 450 W Xe-lamp (Edinburgh Instruments, Livingston, UK). The decay curves were recorded with a 60 W microsecond flashlamp (Edinburgh Instruments, Livingston, UK).

## 3. Results and Discussion

### 3.1. Crystal Structures

Figure 1a shows the XRD patterns of KYC:3%Dy^3+^,5%Ce^3+^, KYC:5%Ce^3+^, and KYC:3%Dy^3+^. Three diffraction spectra of the sample show the similar diffraction peaks with slight differences in intensities. All diffraction peaks match well with the standard diffraction spectrum of KHo(CO_3_)_2_ (JCPDS: 1-88-1419). No secondary phases are detected in the spectrum. The crystal structure is the same as reported in the literature [24]. In the enlarged image, 2*θ* angles at the (002) crystal plane are observed to shift to a lower angle in comparison to KHo(CO_3_)_2_. The reason for lattice expansion is that the radius of doped ions is larger than that of Ho^3+^ ions. The ionic radii of Ho^3+^, Y^3+^, Dy^3+^, and Ce^3+^ are 1.015, 1.019, 1.027, and 1.143 Å [26], respectively. Figure 1b presents the Rietveld refinement of KYC:3%Dy^3+^,5%Ce^3+^ with FullProf software. KHo(CO_3_)_2_ was selected as the initial structural mode. The refined cell parameters are presented in Table 1. From the refined lattice parameters of KYC:3%Dy^3+^,5%Ce^3+^, the values of the cell volume and parameters are larger than those of KY(CO_3_)_2_, which indicates that Ce^3+^ and Dy^3+^ ions were successfully doped into the monoclinic structure of KY(CO_3_)_2_ [27].

### 3.2. Morphologies and Element Analysis

The morphologies of KYC:3%Dy^3+^,*y*%Ce^3+^ (*y* = 0, 1, 2, 3, 4, 5, 6, 7) are illustrated in Figure 2a–h. According to the scanning electron microscopy images, both KYC hosts doped with Dy^3+^ and Ce^3+^ are monoclinic and consist of well-crystallized grains. In Figure 2a, for KYC:3%Dy^3+^, the grain size is relatively uniform, ranging from 20 to 50 microns. However, when Ce^3+^ ions were added to KYC:3%Dy^3+^, the grains grew and became more likely to fracture. Many small particle fragments were found in the middle of the grains (Figure 2b–h). This fragmentation of grains may be caused by excessive internal stress in the grains due to the significant difference in ionic radii between Dy^3+^ and Ce^3+^. Ce^3+^ and Dy^3+^ ions were doped into KYC through a chemical reaction that generates different small crystal nuclei for Ce^3+^, Dy^3+^, and Y^3+^. The crystals merge and grow, but due to differences in crystal growth direction between the various small crystal nuclei, it is impossible for the crystals to form a uniform size completely. As the concentration of Ce^3+^ doping increases, primary grains that have not been completely merged can still be found at the fracture of bulk grains. The presence of pores and spaces between grains makes them more prone to fragmentation.

In addition, Figure 2i shows the elemental analysis of KYC:3%Dy^3+^,5%Ce^3+^ examined by energy dispersive spectroscopy. Although the doping concentrations of Ce^3+^ and Dy^3+^ are relatively low, it can still be observed from the energy spectrum that the grain contains elements such as K, Y, Ce, and Dy. The element contents of Y, Ce, and Dy are 9.9%, 0.4% and 0.3%, respectively. The ratio of Y to the total content of Ce and Dy is 14.1, which is close to the original stoichiometric ratio of 13.3. These results indicate that Ce^3+^ and Dy^3+^ have been successfully doped into the KYC host.

### 3.3. Luminescent Properties

Figure 3a displays the typical emission spectra of KYC:*x*%Dy^3+^ when exposed to an excitation wavelength of 365 nm. The emission spectrum of Dy^3+^ has four distinct emission bands, with peak values located at 492, 577, 666, and 756 nm. These peaks correspond to the transition from ^4^F_9/2_ to ^6^H_15/2_, ^6^H_13/2_, ^6^H_11/2_, and ^6^H_9/2_ [14,28], respectively. The emission intensity initially increases and then decreases with the doping concentration of Dy^3+^. The maximum emission intensity occurs at a doping concentration of 3% Dy^3+^; excessive Dy^3+^ ions will reduce the distance between two adjacent Dy^3+^ ions, resulting in concentration quenching [16,17,18]. The reason is that cross-relaxation consumes the number of excited state electrons at the emission level of ^4^F_9/2_ between the two similar energy gaps of ^4^F_9/2_-^6^F_3/2_ and ^6^H_9/2_-^6^H_15/2_. This relaxation process can be expressed as follows: ^4^F_9/2_+^6^H_15/2_ → ^6^F_3/2_+^6^H_9/2_. Additionally, the emission of excited-state electrons in Dy^3+^ mainly concentrates in the regions 462–504 nm and 553–597 nm. The emission colors are blue and yellow for the two main emission bands. To investigate the emission intensity, the emission intensities in the blue and yellow regions are integrated, and the relative intensities of the two emission bands are shown in Figure 3b. The ratios of blue-to-yellow light change with concentration and can be fitted linearly. The fitting results show that the integrated intensity ratio of blue and yellow light is close to 1, and the slope of the fitted line approaches 0. In the Dy^3+^ emission spectrum, the emission of each spectral component remains stable, and the change in doping concentration does not induce hypersensitive transitions of the ^4^F_9/2_ → ^6^H_13/2_ [29]. The normal emission of yellow light indicates that Dy^3+^ is in a relatively stable crystal field environment. The [DyO_8_] polyhedra in the KYC:Dy^3+^ lattice effectively weaken the influence of external electric fields on Dy^3+^ [24]. The energy transfers of adjacent Dy^3+^ between different polyhedra only occur through the zig-zag chains along the C-axis direction [25], which means that Dy^3+^ is less affected by the other electric field environment. Hence, the emission of electronic transitions between different energy levels is relatively stable. Dy^3+^ ions with different concentrations emit a single luminescent color. The luminescent properties of Dy^3+^ ions can be used for color modulation, especially for solid-state lighting. Multi-color luminescence can be achieved simply through spectral adjustment. Finally, 3% of Dy^3+^ was used to synthesize white light in the KYC host.

Figure 4 clearly displays the excitation spectra of KYC doped with Ce^3+^ and Dy^3+^ monitored at 577 nm. Sharp excitation peaks are visible in the excitation spectrum of KYC:3%Dy^3+^. These peaks are observed at 325, 339, 351, 365, 386, 428, 453, and 476 nm and correspond to the transition of Dy^3+^ from ^6^H_15/2_ to ^6^P_3/2_, ^4^I_9/2_, ^6^P_7/2_, ^6^P_5/2_, ^4^I_13/2_, ^4^G_11/2_, ^4^I_15/2_, and ^4^F_9/2_ [5,28], respectively. When Ce^3+^ was added to KYC:3%Dy^3+^, two broad absorption peaks were detected at 274 and 340 nm (monitoring at 577 nm), which belong to the energy transition of Ce^3+^ from 4f to 5d [30]. The excitation intensity of KYC:3%Dy^3+^,5%Ce^3+^ in the range of 230–360 nm is significantly greater than that of KYC:3%Dy^3+^. This is primarily due to the overlap of energy levels between Ce^3+^ and Dy^3+^, which promotes electron resonance migration [31,32]. The other excitation intensities of Dy^3+^ at 428, 453, and 476 nm remain the same as before. Therefore, a higher excitation intensity benefits the emission of Dy^3+^ ions in the resonance excitation wavelength.

Figure 5 presents the emission spectra of KYC:3%Dy^3+^,*y*%Ce^3+^. In Figure 5a, the emission spectrum is mainly composed of the emission of Ce^3+^ and Dy^3+^. The emission wavelength of Ce^3+^ is distributed in the range of 350–460 nm, while Dy^3+^ is in the range of 460–800 nm. When KYC:3%Dy^3+^,*y*%Ce^3+^ is excited at 339 nm, the collaboration between Ce^3+^ and Dy^3+^ enables it to emit a higher fluorescence intensity. As shown in the figure, the emission intensity of Ce^3+^ is maximum when the Ce^3+^ concentration in KYC:3%Dy^3+^,*y*%Ce^3+^ is 4, while the maximum emission intensity of Dy^3+^ corresponds to a Ce^3+^ concentration of 5. Before a concentration of 5%, Ce^3+^ ions enhance the emission intensity of Dy^3+^. However, as the Ce^3+^ ion content continues to increase, the concentration quenching of Ce^3+^ weakens the emission intensity of Dy^3+^ [33]. This phenomenon is mainly caused by the electric multipole interaction between Ce^3+^ ions. The type of electric multipole interaction can be determined using Dexter’s formula [34]:$$\frac{I}{\chi }=K{\left[1+\beta {\left(\chi \right)}^{\frac{\theta }{3}}\right]}^{-1}$$
where *I* is the emission intensity of Ce^3+^ at 399 nm, χ is an activator concentration of Ce^3+^ in KYC:Dy^3+^, Ce^3+^, *θ* is a multipole–multipole interaction type, and *K* and *β* are constants. To obtain the value of *θ*, the above equation can be simplified as follows:$$log\frac{I}{\chi }=-\frac{\theta }{3}log\chi +C$$

After fitting the Ce^3+^ emission data, it was found that the value of *θ* is close to 8. The result shows that the mechanism of Ce^3+^ concentration quenching is attributed to the dipole–quadrupole.

The emission intensities of Dy^3+^ in the presence of Ce^3+^ ions were calculated in Figure 5b. The integral intensity ratios of blue to yellow are similar in KYC:3%Dy^3+^,*y*%Ce^3+^ (*y* = 0–7). The intensity ratio of each sample can be linearly fitted, and the slope of the fitted line is close to 0. The similar intensity ratio of Dy^3+^ indicates that its emission has relative independence. The doping of Ce^3+^ ions only enhances the emission of Dy^3+^ through energy transfer and does not trigger the hypersensitive transition. The relative intensities at different *y* concentrations are shown in Figure 6. The enhancement values are described by I*_y_*/I_0_ and I*_y_*/I_1_ at the emission wavelength of 577 nm, where I*_y_* and I_0_ are the peak intensities excited by 339 nm at KYC:3%Dy^3+^,*y*%Ce^3+^, and KYC:3%Dy^3+^,0Ce^3+^, I_1_ is the maximum peak intensity excited by 365 nm in KYC:3%Dy^3+^. From the curves, the maximum value of enhancement occurs at a Ce^3+^ concentration of 5%, which is 8.43 and 2.32 times for I_0_ and I_1_, respectively. Therefore, Ce^3+^ is employed to not only increase the emission intensity of Dy^3+^ but also to adjust spectral components.

### 3.4. Decay Curves

Figure 7 illustrates the typical photoluminescence decay curve of KYC:3%Dy^3+^,5%Ce^3+^. The phosphor was excited at 339 nm and emitted at 577 nm. From the figure, the decay curve can be fitted by a double-exponential function. The fitted residuals are well distributed around 0. The double-exponential fitting curve indicates that two main factors are dominating the radiation process of Dy^3+^. In general, the lifetimes vary with different concentrations due to the energy transfer between the Dy^3+^ ions [35]. Therefore, a constant concentration of Dy^3+^ was used to investigate the contribution of Ce^3+^. The lifetimes of Dy^3+^ with different Ce^3+^ concentrations (1% to 7%) were calculated. Table 2 provides the detailed values of lifetime for τ_1_ and τ_2_ corresponding to the double-exponential components. From the table, it can be observed that the τ_1_ and τ_2_ values increase with the increase in the Ce^3+^ concentration in the range of 1% to 5%, which is consistent with the trend of emission intensity variation. The increase in lifetime is mainly attributed to the transfer of excited state electrons from Ce^3+^ to Dy^3+^. Overlapping excitation bands are more prone to resonance transfer and cross-relaxation [14,31]. However, as the concentration of Ce^3+^ continues to increase, the number of excited electrons in Dy^3+^ decreases due to the quenching of the Ce^3+^ concentration [33]. As a result, the lifetimes of Dy^3+^ slightly decrease.

### 3.5. Energy Level Diagram

Figure 8 shows the energy transfer mechanism of KYC:Dy^3+^,Ce^3+^. When excited by 339 nm, the electrons of both Ce^3+^ and Dy^3+^ are stimulated from the ground state to the excited state simultaneously. In the Ce^3+^ excited state of ^5^D_3/2_, one of the electrons relaxes to ^2^F_5/2_ and ^2^F_7/2_ producing the emission of 375 and 396 nm through the lowest 5d excited state [31]. The other excited state electrons of Ce^3+^ (^5^D_3/2_) could transfer to the excited energy level of Dy^3+^ (^4^I_9/2_) by the resonance energy transfer (RET) [14,31]. For the excited state of Dy^3+^, the excited electrons initially relax to the lower excited state of ^4^F_9/2_ by non-radiation and then emit blue (^6^H_15/2_), yellow (^6^H_13/2_), and red (^6^H_11/2_, ^6^H_9/2_) light when returning to the ground state [36]. This simultaneous excitation of Ce^3+^ at 339 nm enhances the emission intensity of Dy^3+^. In addition, according to the emission spectra of KYC:3%Dy^3+^,4%Ce^3+^ and KYC:3%Dy^3+^,5%Ce^3+^, there were no simultaneous increases or decreases in the emission intensity of Ce^3+^ and Dy^3+^ ions, indicating the existence of other energy transfer forms. The energy levels of ^6^P_5/2_ (Dy^3+^) and ^4^K_17/2_ (Dy^3+^) in an excited state are 27,503 and 26,365 cm^−1^ [37]. Due to the overlap of Ce^3+^ emission and Dy^3+^ excitation, the energy of the Ce^3+^ excited state electrons can be transported from Ce^3+^ to Dy^3+^ by the cross-relaxation (CR). The cross-relaxation processes can be described as ^5^D_3/2_+^6^H_15/2_ → ^2^F_5/2_+^6^P_5/2_ and ^5^D_3/2_+^6^H_15/2_ → ^2^F_7/2_+^4^K_17/2_. When the concentration of Ce^3+^ exceeds 5%, the emission intensities of Ce^3+^ and Dy^3+^ decrease simultaneously due to the concentration quenching of Ce^3+^.

### 3.6. Chromaticity Coordinates

The Dy^3+^ emission spectrum comprises blue, yellow, and red components. In the KYC:3%Dy^3+^,*y*%Ce^3+^ spectrum, the emission intensity of yellow light is consistently similar to that of blue light at different Ce^3+^ concentrations. Therefore, the luminescence color can be adjusted by changing the concentration of Ce^3+^ ions. Figure 9 shows the chromaticity coordinates of KYC:3%Dy^3+^,*y*%Ce^3+^ (*y* = 0–7) and KYC:5%Ce^3+^. From the graph, it is evident that KYC:3%Dy^3+^ emits light in the yellow-green region, KYC:5%Ce^3+^ emits light in the purplish-blue region, and all Ce-doped KYC:3%Dy^3+^,*y*%Ce^3+^ (*y* = 1–7) color coordinates appear in the white light region. The KYC:Dy^3+^ phosphor is used for white light mainly due to the relatively consistent and high emission intensity of blue and yellow light in the Dy^3+^ emission spectrum. Moreover, KYC:Dy^3+^ luminescent color is close to daylight, and its luminescent color is similar to the emission of Dy^3+^ in Bi(PO_4_)_6_ [3], Y_4_AI_2_O_5_ [5], and K_3_YB_6_O_12_ [16], belonging to warm white light. It is easy to adjust the luminescent color by adding other luminescent ions to the KYC:Dy^3+^, while the luminescent color of Dy^3+^ remains constant.

The correlated color temperatures were calculated by the formula [38], and the detailed values are provided in Table 3. The coordinate of KYC:3%Dy^3+^,5%Ce^3+^ is (0.3204, 0.2972) with a correlated color temperature of 6270 K, which is closer to standard daylight (0.33, 0.33). Moreover, the other Ce-doped KYCs emit cold white light. These cold white lights have relatively high color temperatures and can be used in locations that require clear vision, such as factories, exhibition rooms, conference rooms, or product exhibitions.

## 4. Conclusions

Monoclinic KYC:Dy^3+^,Ce^3+^ was successfully prepared by the hydrothermal method. In the KYC host, Dy^3+^ maintains relatively independent emission characteristics. The relative intensity of blue and yellow light in the emission spectrum is not affected by the doping concentration of Dy^3+^ ions. Even when Ce^3+^ and Dy^3+^ are co-doped with KYC, Dy^3+^ can still maintain relatively independent emission. The relative intensity ratio of blue-to-yellow light is a constant value in each sample. The cross-relaxation and resonance energy transfer between Ce^3+^ and Dy^3+^ effectively enhance the emission intensity of Dy^3+^. Moreover, all Ce-doped KYC samples can emit white light. The light emitted by KYC:3%Dy^3+^,5%Ce^3+^ is closest to standard daylight. These results indicate that both Ce^3+^ and Dy^3+^ ions maintain relatively independent emission characteristics in the KYC host, which makes it convenient to tune the emission color.

## Figures and Tables

**Figure 1 materials-17-01438-f001:**
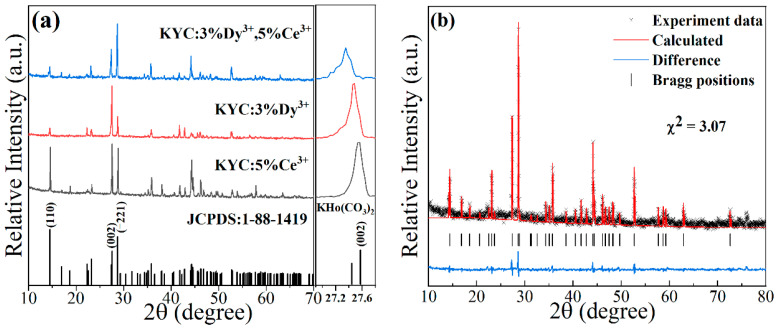
(**a**) XRD patterns of KYC:5%Ce^3+^, KYC:3%Dy^3+^, and KYC:3%Dy^3+^,5%Ce^3+^; (**b**) Rietveld refinements of KYC:3%Dy^3+^,5%Ce^3+^.

**Figure 2 materials-17-01438-f002:**
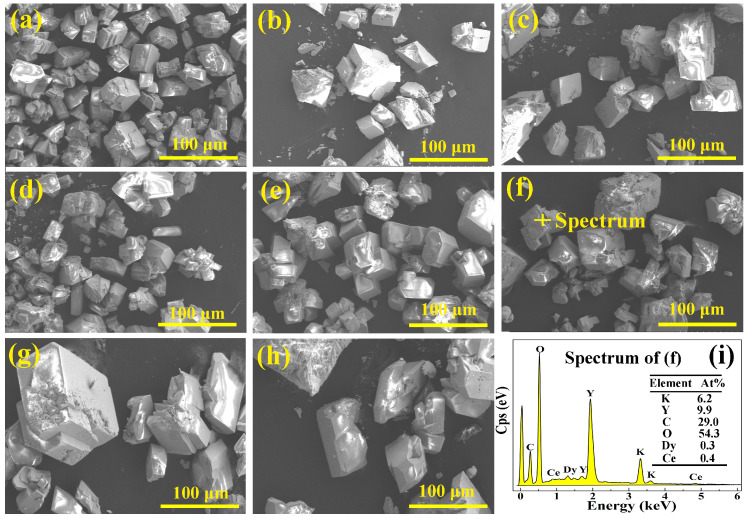
(**a**–**h**) Morphologies of KYC:3%Dy^3+^,*y*%Ce^3+^ (*y* = 0, 1, 2, 3, 4, 5, 6, 7); (**i**) energy dispersive spectrum.

**Figure 3 materials-17-01438-f003:**
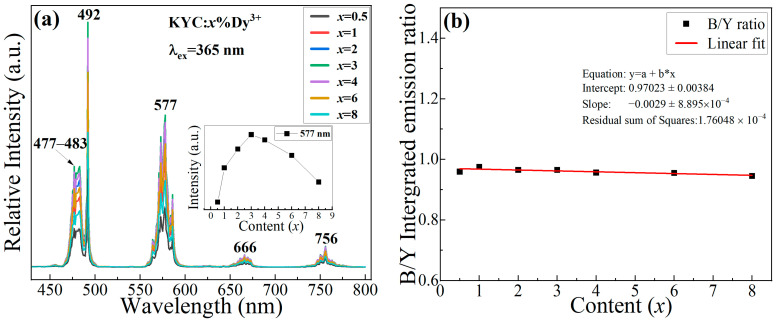
(**a**) Emission spectra of Dy^3+^-doped KYC; (**b**) integrated intensity ratio of blue to yellow (B/Y).

**Figure 4 materials-17-01438-f004:**
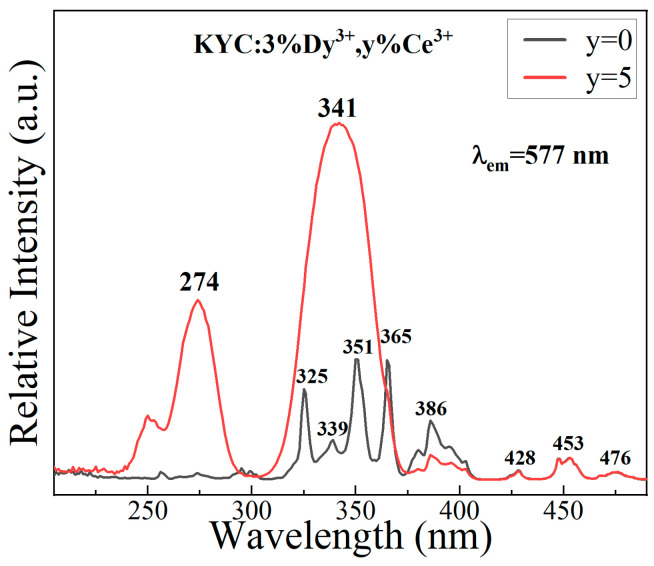
Excitation spectra of KYC:3%Dy^3+^,*y*%Ce^3+^ (*y* = 0, 5).

**Figure 5 materials-17-01438-f005:**
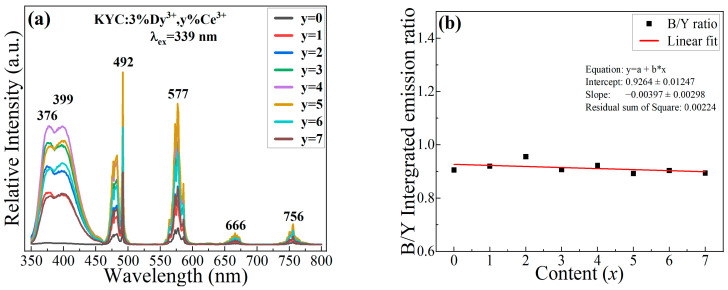
(**a**) Emission spectra of KYC:3%Dy^3+^,*y*%Ce^3+^; (**b**) integrated intensity ratio of blue to yellow (B/Y).

**Figure 6 materials-17-01438-f006:**
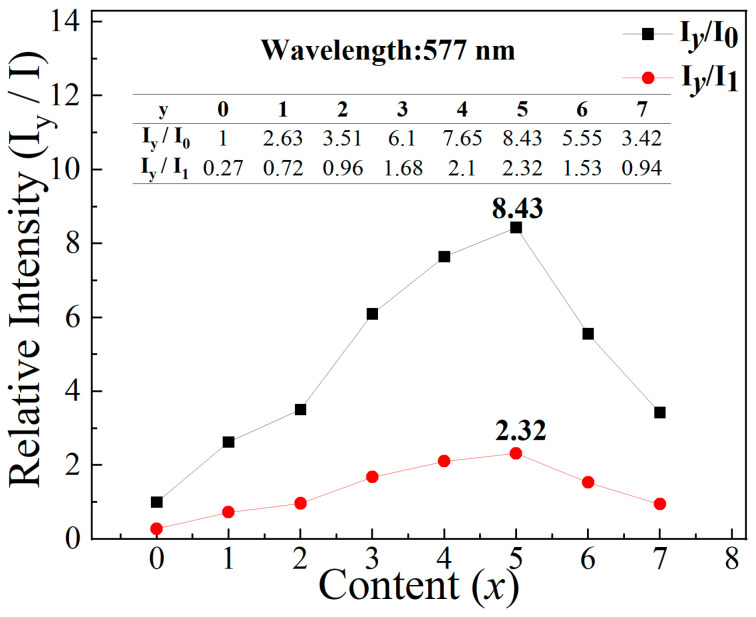
Relative intensities of KYC:3%Dy^3+^,*y*%Ce^3+^ at different *y* concentrations.

**Figure 7 materials-17-01438-f007:**
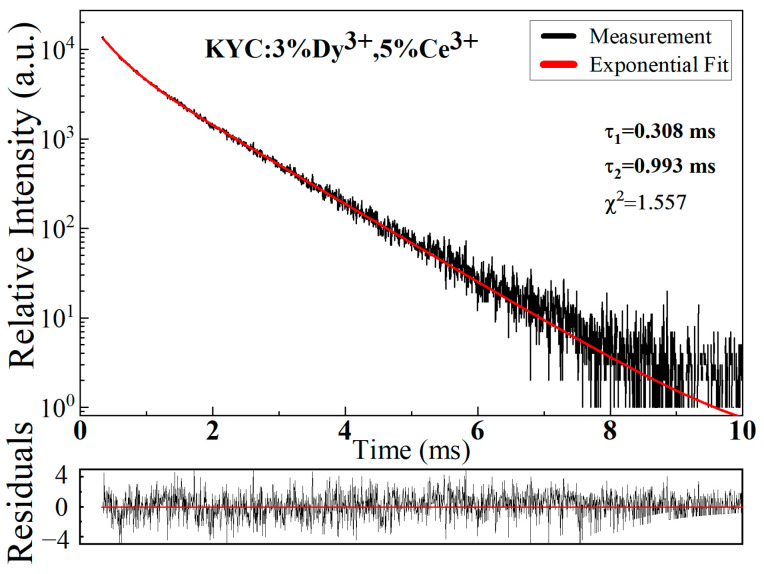
Photoluminescence decay curve of KYC: 3%Dy^3+^,5%Ce^3+^.

**Figure 8 materials-17-01438-f008:**
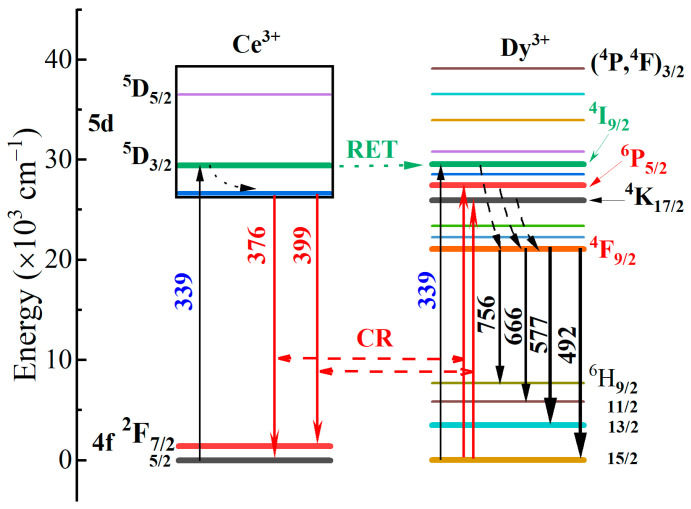
Energy transfer mechanism of Ce^3+^ and Dy^3+^ in KYC:3%Dy^3+^,*y*%Ce^3+^.

**Figure 9 materials-17-01438-f009:**
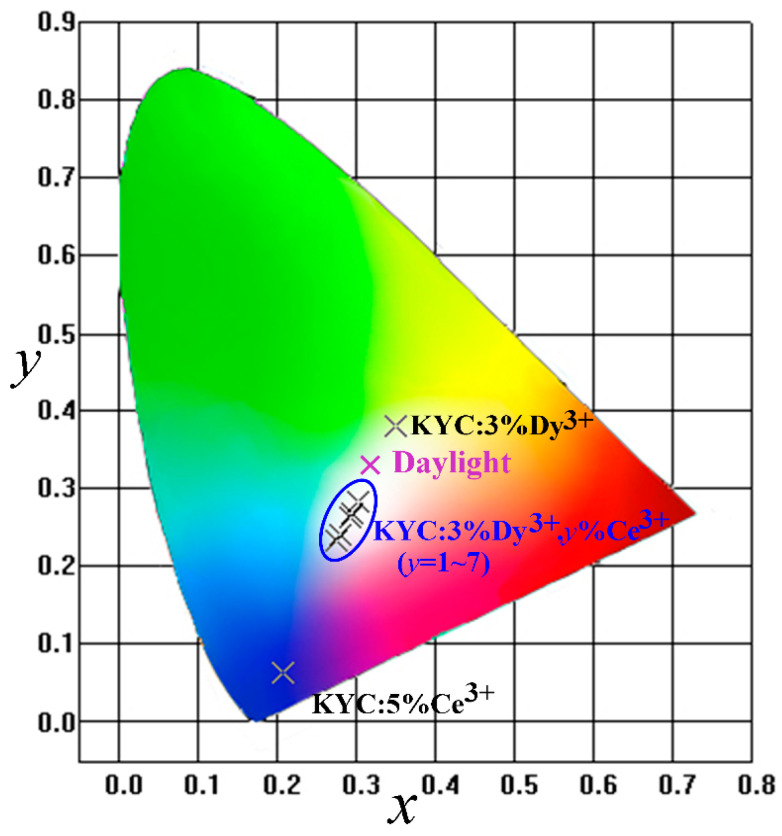
CIE chromaticity coordinates (×) of KYC: 3%Dy^3+^,*y*%Ce^3+^ excited at 339 nm.

**Table 1 materials-17-01438-t001:** Refined parameters of KYC:3%Dy^3+^,5%Ce^3+^.

Parameter	KY(CO_3_)_2_ [24]	KYC:3%Dy^3+^,5%Ce^3+^ [This Work]
Crystal System	Monoclinic	Monoclinic
Space group	C2/c	C2/c
a (Å)	8.488	8.489
b (Å)	9.442	9.447
c (Å)	6.913	6.922
α (°)	90.00	90.00
β (°)	110.963	110.94
γ (°)	90.00	90.00
Cell Volume (Å^3^)	517.4	518.340

**Table 2 materials-17-01438-t002:** Lifetimes of KYC:3%Dy^3+^,*y*%Ce^3+^.

No.	*y*%	τ_1_ (μs)	τ_2_ (μs)	τ (μs)
1	1%	274.3	949.1	826.1
2	2%	277.4	950.2	827.4
3	3%	290.1	974.7	844.5
4	4%	300.2	985.6	859.2
5	5%	308.4	993.3	858.7
6	6%	292.9	974.4	851.0
7	7%	311.5	985.9	850.7

**Table 3 materials-17-01438-t003:** Chromaticity coordinates and correlated color temperatures.

NO.	Samples	*x*	*y*	CCT (K)
1	KYC: 3%Dy^3+^	0.3517	0.3524	4762
2	KYC: 3%Dy^3+^,1%Ce^3+^	0.3038	0.2743	8067
3	KYC:3%Dy^3+^,2%Ce^3+^	0.3034	0.2737	8129
4	KYC: 3%Dy^3+^,3%Ce^3+^	0.3174	0.2919	6527
5	KYC: 3%Dy^3+^,4%Ce^3+^	0.3138	0.2860	6879
6	KYC:3%Dy^3+^,5%Ce^3+^	0.3204	0.2972	6270
7	KYC: 3%Dy^3+^,6%Ce^3+^	0.3149	0.2869	6777
8	KYC: 3%Dy^3+^,7%Ce^3+^	0.3129	0.2840	6984
9	KYC: 5%Ce^3+^	0.2078	0.0632	1759
10	Daylight	0.33	0.33	5616

## Data Availability

Data are contained within the article.

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
