# Peer review of "Fixed Yellow-to-Blue Intensity Ratio of Dy3+ in KY(CO3)2 Host for Emission Color Tuning"

_materials, 2024, doi:10.3390/ma17061438_

Round 1

Reviewer 1 Report

Comments and Suggestions for Authors

I read the article with interest and suppose that it can be published in the journal. However, a revision and some clarification would be required. 
1. An abstract should be rewritten.  The authors use slang, which is explained in the scientific article. 

First sentence.
Co-doped KY(CO3)2 monoclinic phosphors with Dy3+ and Ce3+ were synthesized using the hydrothermal method to create a fixed yellow-to-blue ratio emission.
What does the monoclinic phosphor mean?
It should be corrected, for instance, for crystalline phosphor of monoclynis structural type.

Second sentence
In the [YO8] polyhedron of the KY(CO3)2 crystal lattice, Y3+ is partially replaced by Dy3+ and Ce3+ to form an independent crystal
field environment. 
What does independent crystal field environment mean?

The authors claim the excitation transfer effects between codopants. As a rule, such suggestions are  supported both by spectral measurements and by measurements of the photoluminescence kinetics at specified excitation wavelengths. To prove excitation transfer, these data should be included in the article. 
The article needs a major revision.

Comments on the Quality of English Language

Looks good.

Author Response

Dear Ms. Azura Xie and Reviewer:

Thanks for your and the reviewer’s hard work!

According to the reviewer’s constructive comments, we rewrite the abstract and add the description of experimental parameters. The corresponding revised contents are marked in the paper. The reviewer’ comments and our responses to the comments are as follows:

I read the article with interest and suppose that it can be published in the journal. However, a revision and some clarification would be required.

1. An abstract should be rewritten.

The authors use slang, which is explained in the scientific article.

First sentence: Co-doped KY(CO3)2 monoclinic phosphors with Dy3+ and Ce3+ were synthesized using the hydrothermal method to create a fixed yellow-to-blue ratio emission. What does the monoclinic phosphor mean? It should be corrected, for instance, for crystalline phosphor of monoclinic structural type.

Second sentence: In the [YO8] polyhedron of the KY(CO3)2 crystal lattice, Y3+ is partially replaced by Dy3+ and Ce3+ to form an independent crystal field environment. What does independent crystal field environment mean?

Response:

We express our gratitude to the experts in the field of luminescence for reminding us to describe phenomena in a scientific language. Inappropriate descriptions of statements may lead to misunderstandings. So we improved some of the sentences.

Original:

(1) Co-doped KY(CO3)2 monoclinic phosphors with Dy3+ and Ce3+ were synthesized using the hydrothermal method to create a fixed yellow-to-blue ratio emission.

(2) In the [YO8] polyhedron of the KY(CO3)2 crystal lattice, Y3+ is partially replaced by Dy3+ and Ce3+ to form an independent crystal field environment.

(3) Therefore, a stable crystal environment could be used to achieve more color tuning of light.

Corrected:

(1) Dy3+, Ce3+ co-doped KY(CO3)2 phosphors with a monoclinic structure were synthesized by a hydrothermal method to achieve a fixed yellow-to-blue ratio emission.

(2) The [YO8] polyhedron, consisting of a Y atom and 8 oxygen atoms, forms a relatively independent microstructure within the KY(CO3)2 host. Y3+ ions are partially replaced by Ce3+ or dY3+ ions to construct the [CeO8] or [DyO8] polyhedral fluorescence emission unit.

(3) Therefore, a stable [YO8] polyhedral structure could be used to achieve more color tuning of light.

  1. The authors claim the excitation transfer effects between codopants. As a rule, such suggestions are supported both by spectral measurements and by measurements of the photoluminescence kinetics at specified excitation wavelengths. To prove excitation transfer, these data should be included in the article.

Response:

Thank you for the reviewer's reminder. Energy transfer often occurs between two ions, which can be observed through spectral measurements and fluorescence decay curves. In some paragraphs of the article, we only marked the relevant parameters in the figure, and there was a lack of text description, which may affect the reading experience for readers. We have added relevant parameters and enhanced the completeness of the text description.

Original:

(1) Figure 4 clearly displays the excitation spectra of KYC doped with Ce3+ and Dy3+.

(2) Figure 7 illustrates the typical decay curve of KYC:3%Dy3+,5%Ce3+.

Corrected:

(1) Figure 4 clearly displays the excitation spectra of KYC doped with Ce3+ and Dy3+ monitored at 577 nm.

(2) Figure 7 illustrates the typical decay curve of KYC:3%Dy3+,5%Ce3+. The phosphor was excited at 339 nm and emitted at 577 nm.

Reviewer 2 Report

Comments and Suggestions for Authors

In this paper, the authors synthesized Dy3+/Ce3+ co-doped KY(CO3)2 phosphor by a hydrothermal method, and the optical properties were characterized with changing the content of Dy3+ and Ce3+. They found that the emission could be largely improved by the Ce3+ codoping with Dy3+ in generating a white emission, and the optimal concentration of each dopants were found out. Also, the energy transfer from Ce3+ to Dy3+ was validated using the measurement of decay times. The findings of this work are thought to be valuable for readers who have interest in designing the white emissive phosphor. But, some minor points listed below should be addressed before the publication.

1. There is no information about which sample the SEM pictures in figures (a) to (h) in Figure 2 are for. Include information in the figure caption.

2. The particles in the SEM photo in Figure 2 appear to be aggregates of primary particles. Also, they seem to break easily. For practical applications, a grinding process is necessary. It is also desirable to evaluate the change in particle size according to Ce content based on the size of the primary particle. If possible, it is recommended to perform high-magnification SEM measurements to view the size of primary particles and observe changes.

3. In Figure 2(i), there is only the EDS spectrum. It would be better if the content of each ingredient was indicated.

4. The optimal concentration of each doping material to obtain the best luminescence intensity can be seen in Figure 3(a) and Figure 6. When the doping concentration exceeds a certain level, the decrease in luminescence intensity is well explained by concentration quenching. These explanations are in the text. However, there is no mention of the mechanism of concentration quenching. The concentration quenching mechanism strongly depends on the phosphor matrix. Therefore, it would be better to investigate the concentration quenching mechanism of KYC:xDy,yCe phosphor based on the results obtained in Figures 3 and 6 and insert a brief explanation.

Author Response

Dear Ms. Azura Xie and Reviewer:

Thanks for your and the reviewer’s hard work!

According to the reviewer’s constructive comments, we add the sample information, high-magnification SEM, element content and discussion of concentration quenching mechanism. The corresponding revised contents are marked in the paper. The reviewer’ comments and our responses to the comments are as follows:

In this paper, the authors synthesized Dy3+/Ce3+ co-doped KY(CO3)2 phosphor by a hydrothermal method, and the optical properties were characterized with changing the content of Dy3+ and Ce3+. They found that the emission could be largely improved by the Ce3+ codoping with Dy3+ in generating a white emission, and the optimal concentration of each dopants were found out. Also, the energy transfer from Ce3+ to Dy3+ was validated using the measurement of decay times. The findings of this work are thought to be valuable for readers who have interest in designing the white emissive phosphor. But, some minor points listed below should be addressed before the publication.

  1. There is no information about which sample the SEM pictures in figures (a) to (h) in Figure 2 are for. Include information in the figure caption.

Response:

We express our gratitude to the expert for your careful and patient review of the manuscript. Due to our carelessness, the content information of the sample is missing in Figure 2. We have added sample information in the figure caption. Thanks again.

Original:

Figure 2. (a-h) Morphologies of KYC:3%Dy3+,y%Ce3+; (i) Energy Dispersive Spectrum.

Corrected:

Figure 2. (a-h) Morphologies of KYC:3%Dy3+,y%Ce3+ (y=0, 1, 2, 3, 4, 5, 6, 7); (i) Energy Dispersive Spectrum.

  1. The particles in the SEM photo in Figure 2 appear to be aggregates of primary particles. Also, they seem to break easily. For practical applications, a grinding process is necessary. It is also desirable to evaluate the change in particle size according to Ce content based on the size of the primary particle. If possible, it is recommended to perform high-magnification SEM measurements to view the size of primary particles and observe changes.

Response:

Thanks for your suggestion.

We re-measured the cross-section of the KYC:3%Dy3+,y%Ce3+ (y=1, 2, 3, 4, 5, 6, 7) samples and magnified it at high magnification to observe the changes in primary particles. As the concentration of Ce ions increases, the number of perfect grains is decreasing. Many oriented grains were observed at the fracture section of the grains. Oriented growth brings many pores and spaces to block grains, thereby increasing the chance of grain fracture. The high-magnification images of the sample are as follows:

(detailed information, WORD)

From the microscopic images, it can be seen that the size of primary ions gradually decreases as the Ce content increases. Therefore, we discuss the reasons for grain fracture in main text. Meanwhile, we conducted elemental mapping on the grains and found that the elemental distributions were uniform.

We have added relevant discussions and descriptions in the main text.

Corrected: (line 112-119)

Ce3+ and Dy3+ ions are doped into KYC through a chemical reaction that generates different small crystal nuclei for Ce3+, Dy3+, and Y3+. The crystals merge and grow, but due to differences in crystal growth direction between the various small crystal nuclei, it is impossible for the crystals to form a uniform size completely. As the concentration of Ce3+ doping increases, primary grains that have not been completely merged can still be found at the fracture of bulk grains. Oriented growth brings many pores and spaces to block grains, making them more prone to fragmentation.

  1. In Figure 2(i), there is only the EDS spectrum. It would be better if the content of each ingredient was indicated.

Response:

  Thank you for the reviewer's suggestions. We have added ingredient information to the images (Figure 2i) and provided corresponding descriptions in the main text.

Corrected: (line 123-126)

The element contents of Y, Ce, and Dy are 9.9%, 0.4% and 0.3%, respectively. The ratio of Y to the total content of Ce and Dy is 14.1, which is close to the original stoichiometric ratio of 13.3. These results indicate that Ce3+ and Dy3+ have been successfully doped into the KYC host.

  1. The optimal concentration of each doping material to obtain the best luminescence intensity can be seen in Figure 3(a) and Figure 6. When the doping concentration exceeds a certain level, the decrease in luminescence intensity is well explained by concentration quenching. These explanations are in the text. However, there is no mention of the mechanism of concentration quenching. The concentration quenching mechanism strongly depends on the phosphor matrix. Therefore, it would be better to investigate the concentration quenching mechanism of KYC:xDy,yCe phosphor based on the results obtained in Figures 3 and 6 and insert a brief explanation.

Response:

Thanks for your suggestion. There is indeed no relevant description in the article regarding the concentration quenching mechanism of Ce3+ and Dy3+. We discussed the concentration quenching mechanism and added relevant descriptions in the article.

(1) The concentration quenching mechanism of Dy3+ (line 138-140)

The reason is that cross relaxation consumes the number of excited state electrons at the emission level of 4F9/2 between the two similar energy gaps of 4F9/2-6F3/2 and 6H9/2-6H15/2. This relaxation process can be expressed as: 4F9/2+6H15/2→6F3/2+6H9/2

(2) The concentration quenching mechanism of Ce3+ (line 185-195)

This phenomenon is mainly caused by the electric multipole interaction between Ce3+ ions. The type of electric multipole interaction can be determined using Dexter's formula [34]:

where I is the emission intensity of Ce3+ at 399 nm, c is an activator concentration of Ce3+ in KYC:Dy3+, Ce3+, q is a multipole–multipole interaction type, and K and b are constants. To obtain the value of q, the above equation can be simplified as:

After fitting the Ce3+ emission data, it was found that the value of q is close to 8. The result shows that the mechanism of Ce3+ concentration quenching is attributed to the dipole-quadrupole.

Round 2

Reviewer 1 Report

Comments and Suggestions for Authors

Figure 7 illustrates the typical decay curve of KYC:3%Dy3+,5%Ce3+.....

Does compound decay?

It should be changed as following.

Figure 7 illustrates the typical  photoluminescence decay curve of KYC:3%Dy3+,5%Ce3+...

Caption at Fig/7 has to be changed correspondingly as well.

Author Response

Dear reviewer:

  Thank you so much for your professional guidance. Your suggestions will help us to use standardized language in describing experimental phenomena, which will be extremely beneficial in enhancing our writing skills. We have now revised the corresponding description in the text and look forward to receiving your guidance again in the future.